# Dimethyl sulfide (DMS) climatologies, fluxes, and trends - Part B: Sea-air fluxes

Sankirna D. Joge[1,2], Anoop S. Mahajan[1, *], Shrivardhan Hulswar[1], Christa A. Marandino[3], Martí Galí[4,5], Thomas G. Bell[6], Mingxi Yang[6] and Rafel Simo[4]

[1]Indian Institute of Tropical Meteorology, Pune, India

[2]Savitribai Phule Pune University, Pune, India

[3]Research Division 2-Biogeochemistry, GEOMAR Helmholtz Centre for Ocean Research Kiel, Kiel, Germany

[4]Institut de Ciències del Mar (CSIC), Barcelona, Catalonia, Spain

[5] Barcelona Supercomputing Center (BSC), Barcelona, Spain

[6]Plymouth Marine Laboratory (PML), Plymouth, UK

*Correspondence to: Anoop S. Mahajan (anoop@tropmet.res.in)

**Abstract**. Dimethyl sulfide (DMS) contributes to cloud condensation nuclei (CCN) formation in the marine environment. DMS is ventilated from the ocean to the atmosphere, and in most models, this flux is calculated using seawater DMS concentrations and a sea-air flux parameterization. Here, climatological seawater DMS concentrations from interpolation and parameterization techniques are passed through seven flux parametrizations to estimate the DMS flux. The seasonal means of calculated fluxes are compared to identify differences in absolute values and spatial distribution, which show large differences depending on the flux parameterization used. In situ flux observations were used to validate the estimated fluxes from all seven parameterizations. Even though we see a correlation between the estimated and observation values, all methods underestimate the fluxes in the higher range ($> 20$ µmol m$^{-2}$ d$^{-1}$) and overestimate the fluxes in the lower range ($< 20$ µmol m$^{-2}$ d$^{-1}$). The estimated uncertainty in DMS fluxes is driven by the uncertainty in seawater DMS concentrations in some regions but by the choice of flux parameterization in others. We show that the resultant flux is hence highly sensitive to both and suggest that there needs to be an improvement in the estimation methods of global seawater DMS concentration and sea-air fluxes for accurately modeling the effect of DMS on the atmosphere.

## 1 Introduction

Dimethyl Sulfide (DMS) is a volatile organic compound obtained from its precursor dimethylsulfoniopropionate (DMSP) through enzymatic cleavage (Andreae and Crutzen, 1997; Charlson et al., 1987; Simó, 2001; Yang et al., 2014; Abbatt et al., 2019; Galí and Simó, 2015). In seawater, DMS further undergoes biotic and abiotic processes. It is consumed by three major processes: (1) bacterial decomposition, (2) photolysis and, (3) ventilation to the atmosphere (Del Valle et al., 2009; Xu et al., 2019; Zhai et al., 2020). The last process is important as DMS in the atmosphere contributes to the formation of cloud

condensation nuclei (CCN). Once DMS is released into the atmosphere from the sea surface, it is oxidized by hydroxyl radicals (OH), nitrate radical ($NO_3$) and halogen radicals (Br and Cl) to form sulfur dioxide ($SO_2$), methane sulfonic acid, and gas-phase sulphuric acid, which contribute to the formation of CCN (Andreae and Barnard, 1984; Woodhouse et al., 2010; Pazmiño et al., 2005). Hence, DMS has importance in cloud formation, and affects the climate due to its direct and indirect effect on radiative forcing (Yoch, 2002), although some uncertainties remain about its overall impacts and climate feedback (Quinn and Bates, 2011; Quinn et al., 2017).

Although the oceans are the major source of global DMS emissions, minor amounts of DMS has also been found to be emitted from vegetation on land (Vettikkat et al., 2020; Jardine et al., 2015; Yi et al., 2008). However, DMS emitted from the surface ocean is responsible for up to 70 % of the natural sulfur emissions into the global atmosphere (Andreae and Raemdonck, 1983; Carpenter et al., 2012; Hulswar et al., 2022). Considering this, it is important to develop a precise emission inventory for the assessment of climate impacts due to DMS emissions (Mahajan et al., 2015; Yang et al., 2015, 2017; Jin et al., 2018).

The emission of DMS occurs due to differences in concentrations of DMS in the seawater and the atmosphere. The sea-air gas transfer is a complex process, with the wind proven to be one of the most influencing factors (Jahne et al., 1979; Frew et al., 2004; D'Asaro and McNeil, 2008; Blomquist et al., 2017). For example, DMS flux measurements have revealed a decrease in gas transfer at medium to high wind speeds ( $> 10$ m s$^{-1}$), attributed to wave-wind interactions and surfactant effects (Zavarsky et al., 2018), factors typically overlooked in traditional approaches (Bell et al., 2017).Hence, the sea-air gas transfer is parameterized as a function of wind speed. In an earlier comparison, Kettle and Andreae (Kettle and Andreae, 2000) compared three parametrizations viz., Liss & Merlivat (1986), Wanninkhof (1992), and Erickson (1993). They concluded that uncertainty in the flux parameterizations leads to uncertainties in estimating the global DMS flux. Furthermore, different datasets for wind speed, sea surface temperature (SST), and sea surface DMS concentration resulted in relatively small variations in these calculated fluxes ($\leq 25$ %) (Kettle and Andreae, 2000).

Here, we compare global sea-air DMS fluxes derived using seven different gas transfer velocity parameterizations using wind speed and SST. The comparison is conducted using different seawater DMS estimations to identify whether the uncertainty in the emissions is larger because of the uncertainty in seawater DMS concentrations or the flux parameterization. We use one interpolation-based seawater DMS concentration climatology ((Hulswar et al., 2022), hereafter referred to as H22) and two parameterization-based seawater DMS climatologies (Galí et al. (2018), hereafter referred to as G18 and Wang et al. (2020), hereafter referred to as W20). A comparison between the three seawater DMS climatologies is presented in the sister paper (Joge et al., referred to as Joge: Part A). The comparison shows that there is a large difference between the interpolation and proxy-based parameterization methods of estimating seawater DMS concentrations, with the interpolation-based method predicting higher values. Interestingly, both methods show an increase in DMS emissions over the last two decades. Here, we inter-compared the DMS fluxes estimated using seven sea-air flux parameterizations and in situ DMS fluxes and identified the drivers of their uncertainties.

## 2 Data and methodology

For DMS flux calculation, seven parametrization schemes (LM86 (Liss and Merlivat, 1986), E93 (Erickson, 1993), N00a, N00b (Nightingale et al., 2000), Ho06 (Ho et al., 2006), GM12 (Goddijn-Murphy et al., 2012), W14 (Wanninkhof, 2014)) are used with the seawater DMS climatological data of H22, G18, and W20 (please check Joge: Part A for a comparison between the seawater DMS estimations). Each flux parametrization scheme uses wind speed, and some also use SST to estimate the DMS sea-air flux. Wind speed and SST were obtained from the National Centers for Environmental Prediction (NCEP; https://psl.noaa.gov/data/gridded/index.html) (Kalnay et al., 1996) and Centennial in situ Observation-Based Estimates (COBE; https://psl.noaa.gov/data/gridded/data.cobe.html) (Ishii et al., 2005), respectively, for the years from 1948 to 2022, and then monthly averaged to calculate the fluxes. The in situ DMS flux observations measured by eddy covariance or gradient flux techniques were obtained from various studies carried out over the global oceans (Table S1). The corresponding locations of flux observation data are shown in Fig. S5.

In general, all the parameterizations we compare in this study depend on wind speed ($u$) and the Schmidt number ($Sc$), which depends on temperature (T). The Schmidt number ($Sc$) is a dimensionless number defined as the ratio of kinematic viscosity ($v$) and molecular diffusivity (D), i.e., $Sc = v/D$ (Liss and Merlivat, 1986). The DMS sea-air flux is determined by using a bulk flux equation $F = k(C_w - C_a/H)$ where $F$ is the calculated DMS flux, $k$ is the gas transfer velocity, and $C_w$ and $C_a$ are the concentrations of the DMS in the seawater and the atmosphere adjacent to the seawater respectively (Wanninkhof, 2014). $H$ is Henry's law solubility for DMS in seawater, which varies with temperature, which is given as $ln H = -3547/T + 12.64$ (Dacey and Wakeham, 1984). Here, $C_a$ and $C_w$ are measured in situ, while $k$ depends on wind speed. $C_w$ is several orders of magnitude higher than $C_a$; hence $C_a/H$ is often ignored (Yan et al., 2023). It should be noted that previous studies have shown that $C_a$ becomes important when the atmospheric boundary layer is shallow and surface concentration is high (Steiner et al., 2006; Steiner and Denman, 2008). The flux parameterization methods give estimates of the $k$ and $Sc$ values, and we follow $F = kC_w$ for DMS flux estimation with all seven flux parametrizations.

As wind is one of the most influential factors affecting gas transfer, most parameterizations have established different wind speed regimes for which different equations estimate the $k$ values (Liss and Merlivat, 1983; Erickson, 1993). The gas transfer velocity $k$ results from the waterside transfer velocity ($k_w$) and airside transfer velocity ($k_a$). For the rarely soluble gas, airside resistance is usually small and neglected, but DMS solubility increases with a decrease in temperature, and hence, air resistance becomes important (Lana et al., 2011; Marandino et al., 2009; Omori et al., 2017). Most parameterizations agree that at wind speeds less than 3.6 m s$^{-1}$, the surface is generally smooth with few waves, known as the 'smooth surface regime.' When the wind speed is above 3.6 m s$^{-1}$ but less than 13 m s$^{-1}$, it is 'rough surface regime,' and more waves can be seen, enhancing the gas transfer. Above 13 m s$^{-1}$ is known as the 'breaking wave regime,' where bubbles are formed along with the waves, dominant increasing the flux as evident from the Heidelberg circular wind tunnel experiments (Jähne et al., 1984; Jahne et al., 1979; Liss and Merlivat, 1986). The different flux parameterizations estimate the k value in those different wind regimes ($u \leq 3.6$: smooth

surface regime, 3.6< $u \leq 13$: rough surface regime, $u > 13$: breaking wave regime), and these wind regimes are also dependent on the Schmidt number ($Sc$) for each parametrization, where Schmidt number depends on temperature (T).

## 2.1 Flux parameterization methods

### 2.1.1 LM86 Flux Parametrization

LM86 formulated the following equations for the three wind regimes, which are defined below following the results of the Heidelberg experiments (Jahne et al., 1979; Jähne et al., 1984) :

$$k_{lm86} = 0.17 \times \left(600/Sc_{lm86}\right)^{2/3} \times u \qquad\qquad (u \leq 3.6) \qquad\qquad (1)$$

$$k_{lm86} = \left(600/Sc_{lm86}\right)^{1/3} \times (2.85 \times u - 10.26) + 0.61 \times \left(600/Sc_{lm86}\right)^{2/3} \qquad (3.6< u \leq 13) \qquad (2)$$

$$k_{lm86} = \left(600/Sc_{lm86}\right)^{1/3} \times (5.9 \times u - 49.91) + 0.61 \times \left(600/Sc_{lm86}\right)^{2/3} \qquad (u > 13) \qquad (3)$$

Here, $u$ is the wind speed in m s$^{-1}$ at 10 m above the sea surface. The $Sc$ is based on the work carried out by Saltzman *et al.* (1993) and the references therein for the temperature range from 5° C to 30° C using:

$$Sc_{lm86} = 2674 - (147.12 \times SST) + (3.726 \times SST^2) - (0.038 \times SST^3) \qquad\qquad (4)$$

### 2.1.2 E93 Flux Parameterization

Erickson (1993) assumed that the sea surface is a mixture of a low-turbulence area (non-whitecap) and a high-turbulence area (whitecap). The gas transfer velocities are obtained from the radon outgassing data obtained during the expedition of Transient Tracers in the Ocean (TTO) and Geochemical Ocean Sections Study (GEOSECS) (Monahan and Spillane, 1984; Kettle and Andreae, 2000). The gas transfer velocities for other species are calculated using the following conversion formula based on wind speed ranges:

$$k_{e93} = k_{R_n} \times \left(Sc_{e93}/Sc_{R_n}\right)^{-2/3} \qquad\qquad (u < 3.6) \qquad\qquad (5)$$

$$k_{e93} = k_{R_n} \times \left(Sc_{e93}/Sc_{R_n}\right)^{-1/3} \qquad\qquad (u \geq 3.6) \qquad\qquad (6)$$

Here, $k_{R_n}$ (Monahan and Spillane, 1984) and $Sc_{R_n}$ are the gas transfer velocity and Schmidt number for radon, respectively, which are given as follows:

$$k_{R_n} = 2.3 + 1.25 \times 10^{-3} \times u^3 \qquad\qquad (u \text{ in m d}^{-1}) \qquad\qquad (7)$$

$$Sc_{e93} = 1911.3 - 113.7 \times SST + 2.9 \times SST^2 - 0.029 \times SST^3 \qquad\qquad (8)$$

$$Sc_{R_n} = 3147.3 - 201.9 \times SST + 5.5 \times SST^2 - 0.055 \times SST^3 \qquad\qquad (9)$$

### 2.1.3 N00a and N00b Flux Parametrization

Dual tracer methods involving the measurements of sulfur hexafluoride $SF_6$ and 3-Helium ($^3He$) were also used to estimate $k$ (Watson et al., 1991). Nightingale et al. (2000) describe the ideal dual tracer combination as the one with one of the tracers being non-volatile, allowing dilution and dispersion corrections to be applied to the volatile tracer to minimize errors while estimating $k$. Due to the absence of such an ideal marine tracer, Nightingale et al. (2000) introduced a novel method of adding metabolically inactive bacterial spores of *Bacillus globigii* var. *Niger* as a conservative tracer to study the gas exchange in the

North Sea (Watson et al., 1991; Nightingale et al., 2000) along with $SF_6$ and $^3He$ dual tracer for comparison. Combining data from other studies in George's Bank (Wanninkhof et al., 1993) and the West Florida shelf (Wanninkhof et al., 1997) with the North Sea data, the N00a parameterization coefficient was given as

$$k_{n00a} = (0.222 \times u^2 + 0.333 \times u) \times \left( Sc_{n00a} / 600 \right)^{-0.5} \tag{10}$$

However, this study exclusively had data from the Northern Atlantic region. Coale et al. (1996) reported $k$ values by using the

130 dual tracer ($SF_6/^3He$) in the equatorial Pacific Ocean, which was then used to upgrade the N00a parameterization to N00b; the upgraded parameterization is given as

$$k_{n00b} = (0.222 \times u^2 \times shape\ parameter + 0.333 \times u) \times \left( Sc_{n00b} / 600 \right)^{-0.5} \tag{11}$$

Here, the *shape parameter* is used to describe variations in wind speed using Weibull Distribution (Waewsak et al., 2011).

### 2.1.4 Ho06 Flux Parameterization

Ho et al. (2006) applied the dual tracer technique to measure the gas transfer velocity with the wind speed ranging from 7–16 m s$^{-1}$. This was done during the Surface Ocean Lower Atmosphere Study (SOLAS) Air-Sea Gas Exchange (SAGE) campaign. The estimation of Ho06 was derived from the SAGE data, and the gas transfer coefficient is given as,

$$k_{ho06} = (0.266 \pm 0.019) \times u^2 \tag{12}$$

### 2.1.5 GM12 Flux Parametrization

Goddijn-Murphy et al. (Goddijn-Murphy et al., 2012) argued that since the wind does not directly affect the gas transfer, it is the turbulence caused due to wind that helps to form bubbles, which increases gas transfer. Hence, the sea-surface roughness is a better parameter to quantify gas transfer. This study used satellite altimetry data to understand the sea surface roughness and measured DMS gas transfer velocity using the eddy covariance flux determination from eight cruises. This resulted in the new GM12 parameterization, which gives gas transfer velocity given as,

$$k_{gm12} = (2.1 \times u - 2.8) \times \left( Sc_{gm12} / 660 \right)^{-0.5} \tag{13}$$

### 2.1.6 W14 Flux Parametrization

Wanninkhof (1992) used the radiocarbon [14]C data from the Red Sea (Cember, 1989) to understand the $CO_2$ gas exchange rates. Based on this, the parametrization was developed using *Sc* number related to the work carried out by Saltzman *et al.* (1993) with the temperature range between 18º C to 25º C. Further, with the help of better quantification of global wind fields and using data with a broader temperature range (-2º C to 40º C), the parametrization developed in 1992 is being upgraded using revised global ocean [14]C inventories and an improved wind speed product (Wanninkhof, 2014). This new parametrization technique is known as W14, which gives a gas transfer velocity equation:

$$k_{w14} = 0.251 \times u^2 \times \left( {Sc_{w14}}/{660} \right)^{-0.5} \tag{14}$$

Here:

$$Sc_{w14} = 2855.7 - 177.63 \times SST + 6.0438 \times SST^2 - 0.11645 \times SST^3 + 0.00094743 \times SST^4 \tag{15}$$

$Sc_{n00a}$, $Sc_{n00b}$ and $Sc_{gm12}$ use the same formulation as that of $Sc_{lm86}$. LM86 shows three linear regions when compariong *k* vs *u*, as defined by Eq.1-3. GM12 shows a linear dependency on the windspeed, while other formulations show a nonlinear dependency (Figure 1a). When the temperature is changed from -2 ºC to 33.2 ºC, then there is am increase and a spread between the k values. This is due to the temperature dependence of *Sc*, which nonlinearly decreases with temperature (Figure 1b). *Sc* is the ratio of kinematic viscosity (*v*) and molecular diffusivity (D) i.e., *Sc* = *v*/D (Liss and Merlivat, 1986). So, as *Sc* decreases with temperature, the molecular diffusion rate increases from higher concentrations (seawater) to lower concentrations (atmosphere) and hence the value of *k* increases with windspeed (Figure 1). Note that even though Ho06 is independent to the *Sc*, there is a small spread in the values of *k* with temperature. This is due to the Ostwald solubility coefficient for DMS according to McGillis et al. (2000) used in Ho06, and this coefficient has a temperature dependence.

### 2.2 Estimation of uncertainties

The total uncertainty in DMS fluxes ($\sigma_{total}$) is calculated using the standard deviations in seawater DMS concentration ($\sigma_{DMS}$), coefficient of parameterization ($\sigma_k$), and wind speed ($\sigma_{wind}$):

$$\sigma_{total} = \sqrt{\sigma_{DMS}^2 + \sigma_k^2 + \sigma_{wind}^2} \tag{16}$$

Here, $\sigma_{DMS}$ is calculated by calculating standard deviation between H22, W20 and G18. This $\sigma_{DMS}$ is used along with N00a parametrization, windspeed and SST data to estimate the standard deviation in the flux, which is shown in monthly and annual $\sigma_{DMS}$ plots (Fig. S3). Next, $\sigma_k$ is calculated by calculating standard deviation between *k* from all seven flux parametrization equations and this $\sigma_k$ is further used along with H22 seawater DMS climatology data, windspeed and SST data to get standard deviation in flux which is shown in the monthly and annual $\sigma_k$ plot (Fig. S4). Similarly, $\sigma_{wind}$ is calculated by calculating

standard deviation between monthly global wind data from the different sources (NCEP Reanalysis 1, NCEP/DOE Reanalysis 2, ECMWF Reanalysis v5 (ERA5)) and it is used along with N00a parametrization, H22 seawater DMS climatology data and SST to calculate standard deviation in flux (plot is not shown however, area weighted global mean is shown in Table 1). In this analysis, N00b is chosen as it has been used for previous DMS studies (Simó and Dachs, 2002; McNabb and Tortell, 2022; Zhang et al., 2021; Zhao et al., 2003, 2024; Lana et al., 2011; Hulswar et al., 2022) for the calculation of fluxes. Finally, $\sigma_{total}$

is obtained using Eq.(16).

## 3 Results

### 3.1 Salient features and seasonal variations

We estimated the seasonal DMS flux using seven different parameterizations and the global seawater DMS data of H22 (Fig.2), G18 (Fig.S1), and W20 (Fig.S2) climatologies to study the geographical and seasonal variations and the differences between

185 the parameterizations.

Overall, the fluxes estimated using all seven parameterizations follow the seawater DMS concentration distribution, with higher values in the southern/northern hemispheres during their respective summers (Fig.2). Elevated levels are also seen in the Indian, Atlantic, and Pacific Oceans in the extra-tropical regions, where elevated wind speed causes higher sea-air fluxes. While the geographical patterns are similar, there is a large difference in the absolute values among the different

parameterizations. When using the G18 or W20 seawater DMS concentrations, the emissions show a similar difference among the different parameterizations, although the absolute values are lower (Fig.S1 and S2).

In December-January-February (DJF), E93 shows a maximum DMS flux of 45.82 µmol m$^{-2}$ d$^{-1}$ in the Weddell Sea region, where the maximum DMS concentration of 18.67 nM is also calculated in H22 (Joge: Part A). For E93, the flux is more uniformly distributed across the Southern Ocean as compared to the other parameterizations (Fig.2). The other

parameterizations also show elevated values in the Southern Ocean, although the range depends on the parameterization used. For example, the E93 parameterization results in the highest values, exceeding 20 µmol m$^{-2}$ d$^{-1}$ throughout the Southern Ocean, while the LM86 parameterization results in peak values less than 10 µmol m$^{-2}$ d$^{-1}$. Further north, in other ocean basins such as the Indian Ocean Ho06, and N00b predict relatively higher fluxes than E93.

During March-April-May (MAM), most parameterizations lead to elevated fluxes in the North Atlantic Ocean, Caribbean Sea,

Baltic Sea, and North Sea, with the DMS flux ranging from 8.71 to 18.73 µmol m$^{-2}$ d$^{-1}$ using the H22 seawater DMS concentrations. Higher fluxes are also calculated on the western coast of the American continent and in the coastal regions of Africa. The gyres in the equatorial Pacific and Indian Oceans also show higher fluxes, although the Northern Atlantic Ocean has higher fluxes than the other ocean basins. Although all the parameterizations show higher values in the northern hemisphere, E93 shows the highest fluxes, and the LM86 parameterization shows the lowest fluxes. In a similar manner, N00b

shows high flux values (13.8 µmol m$^{-2}$ d$^{-1}$) compared to N00a (11.33 µmol m$^{-2}$ d$^{-1}$) in the Caribbean Sea, probably due to the wind correction factor in the N00b parametrization.

The June-July-August (JJA) period shows high values in the upwelling regions off the continental coasts and the equatorial Indian Ocean and Pacific Ocean. During this period, the geographical variation strongly depends on the parameterization chosen. For example, the E93 parameterization mainly shows peaks in the Arctic Ocean and the northern boundaries of the other ocean basins. However, other parameterizations show peaks in the equatorial oceans in addition to the northern latitudes. This difference in variation is driven by the different responses of the parameterizations to winds.

Flux values start increasing in the Southern Ocean during September-October-November (SON). The flux value estimated by Ho06 were the highest during this period (18.40 µmol m$^{-2}$ d$^{-1}$) in the south Atlantic Ocean along coastal areas of South Africa, although the other parameterizations also show an increase in the Southern Ocean except for LM86. A distinct hotspot is also seen in the Indian Ocean region in all estimations such as Ho06 followed by N00a (13.77 µmol m$^{-2}$ d$^{-1}$), N00b (16.75 µmol m$^{-2}$ d$^{-1}$), GM12 (11.97 µmol m$^{-2}$ d$^{-1}$), and W14 (13.84 µmol m$^{-2}$ d$^{-1}$), while LM86 estimated the least (10.66 µmol m$^{-2}$ d$^{-1}$) in the Indian ocean region.

## 3.2 Differences

We calculated the seasonal differences between all the flux parameterizations with respect to the N00b (Fig.3), however DMS-CO2 flux usually uses W14 flux parametrization but we choose N00b as it is used in the recent DMS climatology papers (Zhao et al., 2024; Wang et al., 2020; Hulswar et al., 2022; Lana et al., 2011). Annually, the largest positive difference is seen in the LM86 parameterization, which consistently displays lower values than the N00b parametrization due to linear dependence of windspeed in LM86 and quadratic in N00b (Eq. 1-3 and Eq.10). The largest negative differences in the polar regions are present in the E93 parameterization, which shows that higher values are calculated at those regions than the N00b parameterization. Although Ho06 also shows large negative differences in the polar regions, large positive differences are present in the mid-latitude and coastal regions. These differences can be as much as 100 % in certain regions, showing that the choice of parameterization plays a crucial role in the DMS flux estimates. The largest positive differences are present in N00b - LM86 in all the seasons, while the largest negative differences can be seen with N00b - E93 (Fig 2). This large negative difference is driven by the differences in the high latitude regions where N00b does not show peaks, for example, in the Southern Ocean (Fig.2). In the mid-latitude and the equatorial regions, peaks are present in N00b estimations and hence N00b - E93 shows the largest positive differences as listed in Table S2. Although N00b is upgraded from N00a parameterization, there is no negative difference between the two parametrizations (Fig.3), which indicates that N00b estimates higher flux values than N00a (Fig.2). The maximum positive differences between the two is listed in Table S2 for all seasons. The differences between N00b and Ho06 are primarily negative (Table S2), but the positive differences are also present in the range from 1.5 to 2.37 µmol m$^{-2}$ d$^{-1}$ but lower than N00b - N00a. The difference between N00b and GM12 is positive. Similarly, in the case of N00b - W14, positive differences are present which can be clearly seen from Fig.3. The summary of the maximum positive and negative values of differences in different oceanic regions is given in Table S2 of supplementary text.

## 3.3 Drivers in flux uncertainties

As explained in the methods section, the total uncertainty in DMS fluxes is derived from the uncertainty in the seawater DMS concentrations, parameterization, and wind speed.

Fig.S3 shows the standard deviation in the DMS flux calculated using the standard deviation between climatological seawater DMS concentrations ($\sigma_{DMS}$) of G18, W20, and H22. Here, the sea-air parameterization is kept constant to isolate the effect of the change due to seawater DMS concentrations. The monthly climatological wind speed data (NCEP reanalysis 1) is used for the flux estimation. From Table S3 the maximum $\sigma_{DMS}$ can be seen in December, January and February in South Atlantic Ocean compared to June, July and August months in North Atlantic Ocean and Arabian Sea. Overall, the largest standard deviation in $\sigma_{DMS}$ can be seen in the Southern Ocean (Fig. S3), where the DMS concentrations are the largest. Fig.S4 shows the standard deviation in the DMS flux due to the standard deviation among seven gas transfer velocity coefficients ($\sigma_k$). Here, we keep the seawater DMS concentrations constant (H22), and monthly climatological wind speed data of NCEP reanalysis 1 is used. The maximum $\sigma_k$ can be seen in December, January and February in Weddell Sea region compared to June, July and August months in Indian Ocean region (Table S3 in supplementary text). From Fig.S3 and Fig.S4, it can be compared that $\sigma_k$ is dominant over $\sigma_{DMS}$ in Weddell Sea region as well as across the coast of Antarctic region. Apart from this coastal region in Antarctica, other coastal regions are dominated by $\sigma_{DMS}$.

Further, the standard deviation in the DMS flux is estimated by calculating standard deviation in wind speed ($\sigma_{wind}$) obtained from different sources. The area weighted global mean flux standard deviation due to $\sigma_{wind}$ is much lower than the area weighted global mean flux standard deviation due to $\sigma_{DMS}$ and $\sigma_k$ on monthly and annual scales (Table 1). Also, from Table S3, it can be seen that maximum $\sigma_{wind}$ is less in all the months and on the annual scale compared to $\sigma_{DMS}$ and $\sigma_k$; even though these values are from different oceanic regions. This shows that the total standard deviation of the sea-air DMS flux ($\sigma_{total}$) is dominated by $\sigma_{DMS}$ and $\sigma_k$, with $\sigma_{wind}$ playing a minor role in the total flux uncertainty (Table S3 and Table 1).

The climatological monthly and annual $\sigma_{total}$ is shown in Fig.4. The maximum $\sigma_{total}$ values in different oceanic regions are shown in Table S3. In most of the months, it can be seen that the oceanic regions where $\sigma_{total}$ is maximum, at the same oceanic regions $\sigma_{DMS}$ is maximum while for some of the months $\sigma_k$ is maximum. So, there is big contribution in $\sigma_{total}$ by both $\sigma_{DMS}$ and $\sigma_k$ but for most of the regions $\sigma_{DMS}$ shows primary contribution while $\sigma_{wind}$ has minor contribution. In Fig.4, the regions where the $\sigma_{total}$ is dominated by the variation in seawater DMS concentrations, i.e., $\sigma_{DMS} > \sigma_k$, are indicated by red dots. The regions where the red dots are absent are the ones where the dominant contribution to $\sigma_{total}$ is due to $\sigma_k$. Also, $\sigma_{total}$ in oligotrophic oceans and most of the coastal areas are dominated by $\sigma_{DMS}$. Annually, the $\sigma_{total}$ in the Southern Ocean is dominated by $\sigma_{DMS}$, but the coastal area of Antarctica is dominated by $\sigma_k$. Table 1 also shows the total DMS$_{sulfur}$ flux to the atmosphere according to each month and annually averaged. For most of the year, the total flux from regions where $\sigma_{DMS}$ is greater than $\sigma_k$ is larger. Indeed, the total annual flux of DMS$_{sulfur}$ to the atmosphere is estimated as 22.08 Tg, of which 17.16 Tg is contributed by areas where $\sigma_k < \sigma_{DMS}$. This indicates that on an annual scale, the uncertainty in DMS$_{sulfur}$ emissions is dominated by seawater DMS concentration. However, from Fig.4, the choice of the flux parametrization also contributes a considerable amount of

uncertainty in the coastal areas of Antarctica, which can be seen in November, December, January, and February. Overall, the choice of seawater DMS estimation method has larger influence on sea-air DMS flux than the choice of flux parameterization, which is also corroborated by analysis presented by Bhatti et.al. (2023) and Tesdal et al. (2016).

### 3.4 Comparison with in situ observations

In situ DMS flux data were compared with the co-located DMS flux data estimated from different parameterizations using the H22 (Fig.5), G18 (Fig.S6), and W20 (Fig.S7). The raw in situ data points are localized and inconsistent in terms of temporal and spatial resolution while models provide the average. So, raw in situ flux data points are not comparable with the model flux values calculated with parametrizations. Hence, for the analysis, raw in situ DMS flux data is binned to $1° \times 1°$ resolution grid box for each month and then flux data points within that box is averaged. Due to binning and averaging localized in situ

information may be lost but for the comparison with DMS flux calculated with parametrization models this is the nearest traditional method for comparisons. After this, ordinary least square regression is applied. For reference, raw in situ DMS flux points are shown in the background (Fig.5, S6 and S7). All flux estimates using either of the DMS seawater climatologies, with any of the flux parameterizations, struggle to match the observations.

In most cases, the flux estimations in the lower range ($< 20 \, \mu mol \, m^{-2} \, d^{-1}$) are overestimated, while the values are underestimated

in the higher range ($> 20 \, \mu mol \, m^{-2} \, d^{-1}$). Indeed, in all the cases, a positive intercept in the linear regressions shows that the emissions are overestimated at lower flux values. This would indicate a constant background flux in the estimated emissions, which would overestimate the total $DMS_{sulfur}$ flux to the atmosphere. In contrast, the fact that the flux estimates do not reproduce the higher DMS fluxes indicates that high emission scenarios, which would contribute strongly to new particle formation and growth, are underestimated by the emission estimations. It should be noted that we use monthly seawater DMS

concentration fields as input. Hence, a difference between the observations and estimations is expected, but there is consistent overestimation of model flux for lower range ($0.1 \, \mu mol \, m^{-2} \, d^{-1}$ to $< 20 \, \mu mol \, m^{-2} \, d^{-1}$) in situ flux points and underestimation for higher range ($> 20 \, \mu mol \, m^{-2} \, d^{-1}$ to $43.4 \, \mu mol \, m^{-2} \, d^{-1}$) in situ flux points. The best match in the lower range is found when using the W20 seawater DMS estimations (Fig.S7), although the slope is consistently lower than 0.33, and the intercept is higher than $2.17 \, \mu mol \, m^{-2} \, d^{-1}$ for all the flux parameterizations ($R^2 < 0.32$ for all the parameterizations). Both H22 and W20

perform better than G18, but none of the correlation coefficients are found to be important, and all the flux parametrization methods fail to reproduce the in situ DMS flux values, particularly the high values of fluxes (Fig.5, S6 and S7).

### 4 Discussion

This study has been conducted to quantify the factors that contribute to the total uncertainty in DMS fluxes to the atmosphere.

From our analysis, it was found that the total uncertainty in the DMS fluxes is dominated by the uncertainty in the seawater

DMS concentrations, followed by the coefficient of gas transfer velocity used in flux parametrization equations. The uncertainty due to winds peed is negligible in comparison.

The seawater DMS concentrations estimated by G18, W20 and H22 have large differences between themselves (please check Joge: Part A). This is a major source of uncertainty and shows the need for more detailed long-term observations across different ocean basins. The present available observations are not consistent in terms of temporal and spatial resolution and some regions like the Southern Ocean are highly under sampled but very important due to high DMS emissions. Hence, models do not fully capture the seawater DMS variations, which translate into uncertainty in the emissions.

In addition to seawater DMS observations, which we hope will be undertaken in future, there are some regions where uncertainty in the total DMS flux is mostly due to the $k$ values. From $k$ vs $u$ plots (Fig.1a) of the seven flux parametrization methods, it is seen that there are large differences among these seven methods. In LM86, N00a, N00b, GM12 and W14 there is a spread in the values of $k$ due to the Schmidt number *(Sc)*. This spread arises from the SST. Even though E93 uses this *Sc* number, the spread is smaller compared to other parametrization, while there is negligible spread in Ho06. To calculate the DMS flux at present we do not use the $C_a$ values as we assume that DMS is supersaturated in seawater. However, past studies have shown that in some special cases, such as when the atmospheric boundary layer is shallow on cold nights or in winter, it is important to consider the airside DMS concentration. In one of the model study, it was found that the difference in the emissions on considering $C_a$ can be as high as 50% (Steiner and Denman, 2008; Steiner et al., 2006), which adds to the uncertainty. The $k$ vs $u$ plots (Fig.1a) are comparable between the seven parameterizations and the total uncertainty due to windspeed from different sources is negligible. Like in situ DMS observations, in situ flux observations are important in order to develop more accurate flux parametrizations. Observations collected from the different flux techniques like eddy covariance, gradient flux, etc. can add to the uncertainty in flux observation data and cross-comparison between the methods across a range of fluxes needs to be undertaken.

The gas transfer velocity equation of W14 uses the square of the average neutral stability winds at 10 m height or second moment i.e., average of the quadratic windspeed. In this study, we used monthly average windspeed i.e., the quadratic of average windspeed for W14. The first method of calculation will estimate higher $k$ values than the second one due to the averaging of the winds. We checked the differences between the two and found that the maximum difference is not more than 4.3 cm h$^{-1}$ for June, July and August months and it is less than 2 cm h$^{-1}$ for rest of the year, which does not contribute pointedly to the large uncertainty.

From 1998 to 2010, both G18 and W20 show an increasing trend in seawater DMS (Joge: Part A). Using the calculated seawater DMS concentrations, G18 and W20 DMS flux trends are also calculated for each parameterization method using the bootstrap resampling method (explained in Part A). The DMS flux trend also shows an increase for all the parameterizations (Figures S8 and S9). DMS flux values are between 3 µmol m-2 d-1 and 6 µmol m-2 d-1 for E93 and Ho06, but lower in LM86. GM12 and W14 show a similar range while N00b shows a larger range compared to N00a.

The DMS flux derived from both empirical and prognostic models shows the poor agreement with fluxes from the point observations (Tesdal et al., 2016), which can also be seen with the flux parametrization methods used in this study when

compared with the in situ DMS flux observations (Fig.5). Tesdal et al.(2016) also concluded that there is large uncertainty in the temporal and spatial distribution of DMS concentrations and fluxes. The total sea-air DMS flux depends primarily on global mean surface ocean DMS concentrations, and the spatial distribution of DMS concentration and the magnitude of the gas exchange coefficient are of secondary importance. In our study, it is primarily seawater DMS concentrations that needs to estimated accurately as $\sigma_{DMS}$ dominates over $\sigma_k$ at most of the regions of global ocean but for some regions it is important to consider $\sigma_k$ over $\sigma_{DMS}$, which agrees with the study of Tesdal et al.(2016).

## 5 Conclusions

The sea-air DMS flux was estimated using different seawater DMS climatologies (see Joge: Part A), wind, and SST as input to seven different flux parameterizations. All the flux estimations show a similar seasonal variation, with peaks in the summers of each hemisphere. However, there were large geographical and absolute flux differences among the different estimations, showing that the DMS$_{sulfur}$ flux to the atmosphere is sensitive to the chosen seawater DMS fields and the chosen flux parameterization. The total uncertainty in flux estimation is dominated by the uncertainty in seawater DMS concentrations and the choice of flux parametrization, while the effect on the total uncertainty due to the different sources of wind speed is less important; however, this might not be true when comparing to in situ fluxes as the gustiness of wind might play an important role. In certain parts of the globe, such as the Peru upwelling region, the South Pacific Ocean, Indian Ocean, Arabian Sea, Bay of Bengal, Coastal areas, North Atlantic Ocean, Gulf of Alaska, and Southern Ocean, etc., the differences between the climatological estimated seawater DMS of G18, W20, H22 is seen in the figures (Paper: Part A). Hence, the uncertainty in the total flux emission is dominated by the uncertainty due to the seawater DMS concentration in these areas where the differences are important (Fig.4). In other regions, uncertainty is dominated by the choice in the coefficient of the flux parametrization, such as the coastal area of Antarctica and the Arctic Ocean. A comparison of in situ and co-located estimated flux showed that all the parameterizations overestimate the DMS flux below 20 µmol m$^{-2}$ d$^{-1}$ but underestimate fluxes larger than 20 µmol m$^{-2}$ d$^{-1}$. This suggests that emissions in current models overestimate the total sea-air DMS flux but underestimate the higher range (> 20 µmol m$^{-2}$ d$^{-1}$ to 43.4 µmol m$^{-2}$ d$^{-1}$) when it can impact new particle formation and growth.

**Code availability**

Codes for the analysis and figures are available on request.

**Data availability**

All the data used here are publicly available and links are provided in the manuscript.

**Competing Interests**

The authors declare that they have no conflict of interest.

**Author Contributions**

ASM conceptualized the study. SJ analyzed the data with help from SH. CM, MG, MY, TB and RS helped with the data, ideas and understanding of the study. SJ and ASM wrote the manuscript with the help of all the coauthors.

**Acknowledgements**

The Indian Institute of Tropical Meteorology is funded by the Ministry of Earth Sciences, Government of India. MG and RS acknowledge support from the European Research Council (ERC) under the European Union's Horizon 2020 research and innovation program (grant agreement #834162, SUMMIT Advanced Grant to RS), and the Spanish Government through grant GOOSE (PID2022_140872NB_I00) as well as the "Severo Ochoa Centre of Excellence" accreditation grant CEX2019-000928-S.

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

**Tables :**

**Table 1.** Area Weighted Global mean flux standard deviation for each month and annually due to $\sigma_{DMS}$, $\sigma_k$ and $\sigma_{wind}$. Also, DMS$_{Sulfur}$ emissions for each month and annually from the areas with $\sigma_{DMS} > \sigma_k$ and the area $\sigma_{DMS} < \sigma_k$ and the total emission across the globe is computed using the N00b flux parameterization and H22 DMS climatology.

| Month | Area Weighted Global Mean Flux std. due to $\sigma_{DMS}$ ($\mu$mol m$^{-2}$ d$^{-1}$) | Area Weighted Global Mean Flux std. due to $\sigma_k$ ($\mu$mol m$^{-2}$ d$^{-1}$) | Area Weighted Global Mean Flux std. due to $\sigma_{wind}$ ($\mu$mol m$^{-2}$ d$^{-1}$) | DMS$_{Sulfur}$ emissions where $\sigma_{DMS} > \sigma_k$ (Tg) | DMS$_{Sulfur}$ emissions where $\sigma_{DMS} < \sigma_k$ (Tg) | Total DMS$_{Sulfur}$ emissions (Tg) |
|---|---|---|---|---|---|---|
| January | 1.85 | 1.69 | 0.16 | 1.47 | 0.85 | 2.33 |
| February | 1.42 | 1.29 | 0.13 | 1.07 | 0.68 | 1.74 |
| March | 1.52 | 1.28 | 0.13 | 1.54 | 0.50 | 2.04 |
| April | 1.07 | 0.99 | 0.10 | 0.98 | 0.52 | 1.50 |
| May | 1.31 | 1.09 | 0.11 | 1.11 | 0.51 | 1.62 |
| June | 1.51 | 1.09 | 0.11 | 1.24 | 0.49 | 1.73 |
| July | 1.39 | 1.09 | 0.12 | 1.29 | 0.52 | 1.81 |
| August | 1.41 | 1.08 | 0.12 | 1.42 | 0.47 | 1.89 |
| September | 1.04 | 0.83 | 0.09 | 1.09 | 0.41 | 1.50 |
| October | 1.08 | 0.94 | 0.10 | 0.97 | 0.63 | 1.60 |
| November | 1.79 | 1.47 | 0.14 | 1.36 | 0.60 | 1.96 |
| December | 1.82 | 1.70 | 0.16 | 1.40 | 0.90 | 2.30 |
| **Annual** | **1.44** | **1.21** | **0.12** | **17.16** | **4.93** | **22.08** |

**Figures :**

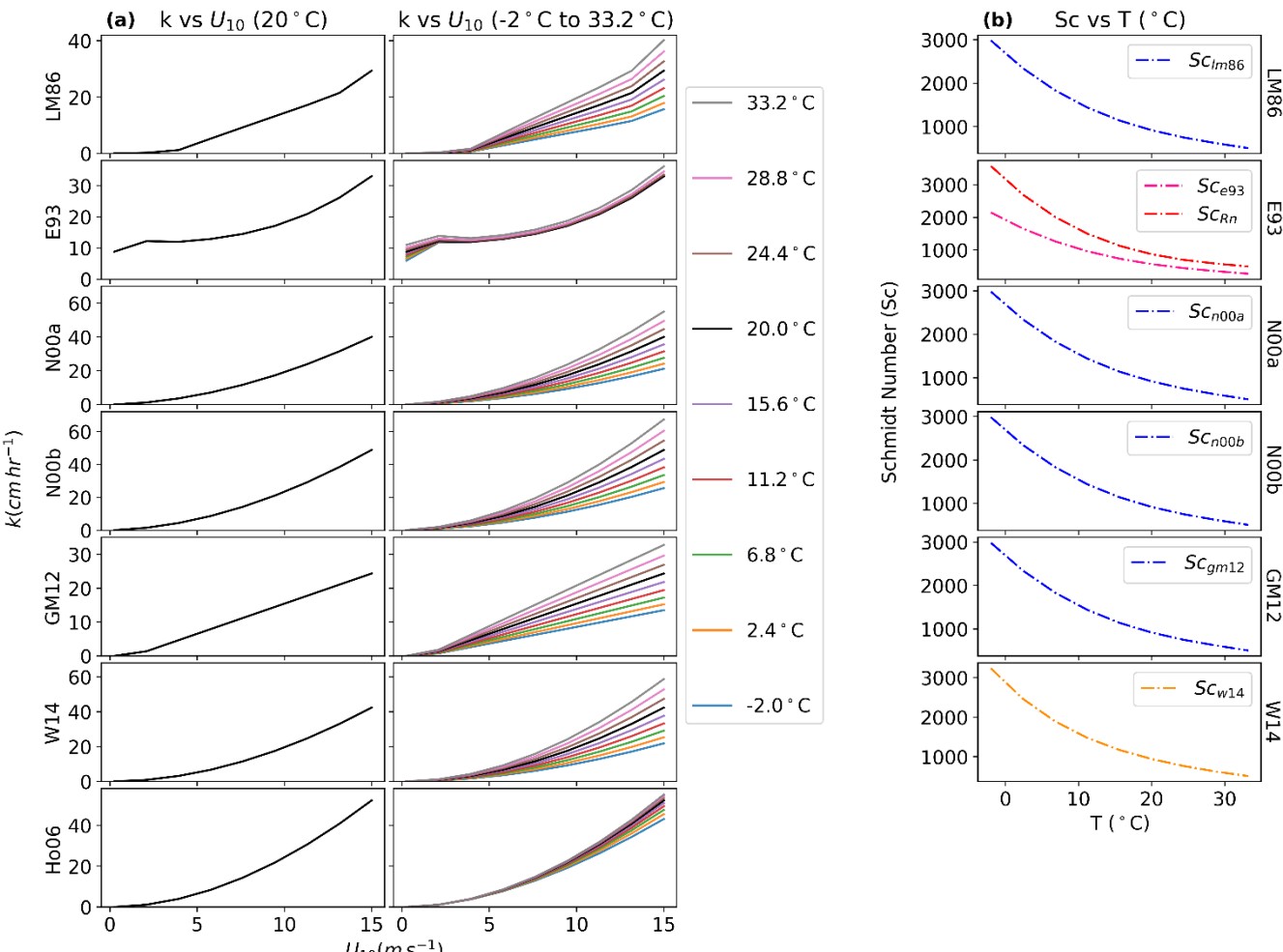

**Figure 1: (a)** Coefficient of gas transfer velocity ($k$) vs windspeed at 10 m from sea surface ($U_{10}$) at constant temperature (20 ºC) and at different temperature values (-2 ºC to 33.2 ºC). **(b)** Schmidt number ($Sc$) vs temperature (T) for each flux parameterization methods. $Sc_{n00a}$, $Sc_{n00b}$ and $Sc_{gm12}$ has same equation as that of $Sc_{lm86}$. $Sc_{Rn}$ is the Schmidt number for radon used in E93 parameterization. Schmidt number ($Sc$) decreases with increase in temperature and hence gas transfer coefficient ($k$) increases.

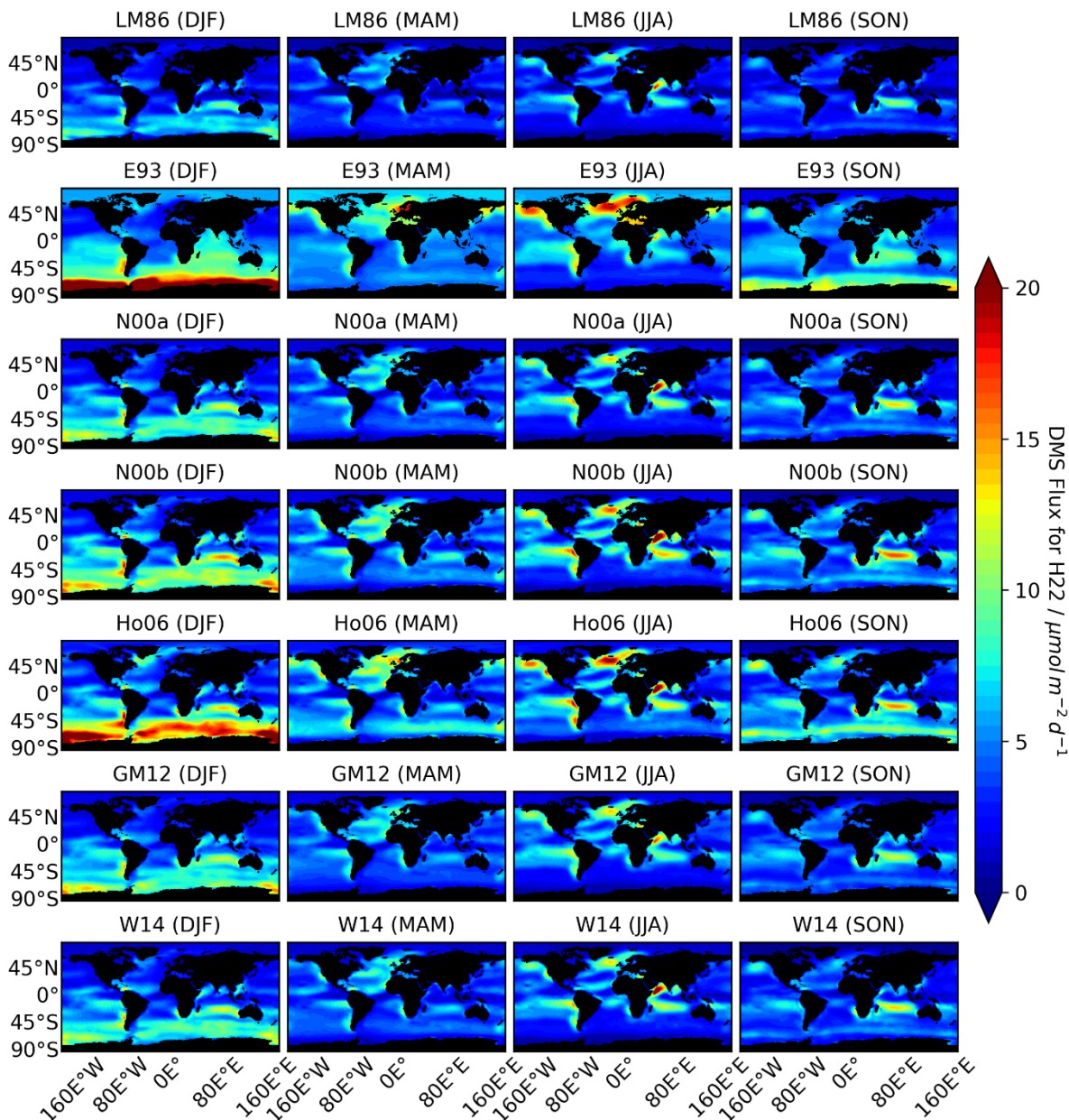

**Figure 2:** DMS fluxes estimated using the seven parameterizations for different seasons using the H22 climatology. The geographical pattern is similar in all the estimates, although the absolute values differ according to the parameterization chosen. In June-July-August (JJA),a maximum flux of 33.75 µmol m$^{-2}$ d$^{-1}$ is calculated in Indian ocean near Somalia with N00b. In December-January-February (DJF),a maximum flux of 45.82 µmol m$^{-2}$ d$^{-1}$ is calculated in Weddell Sea region with E93.

580

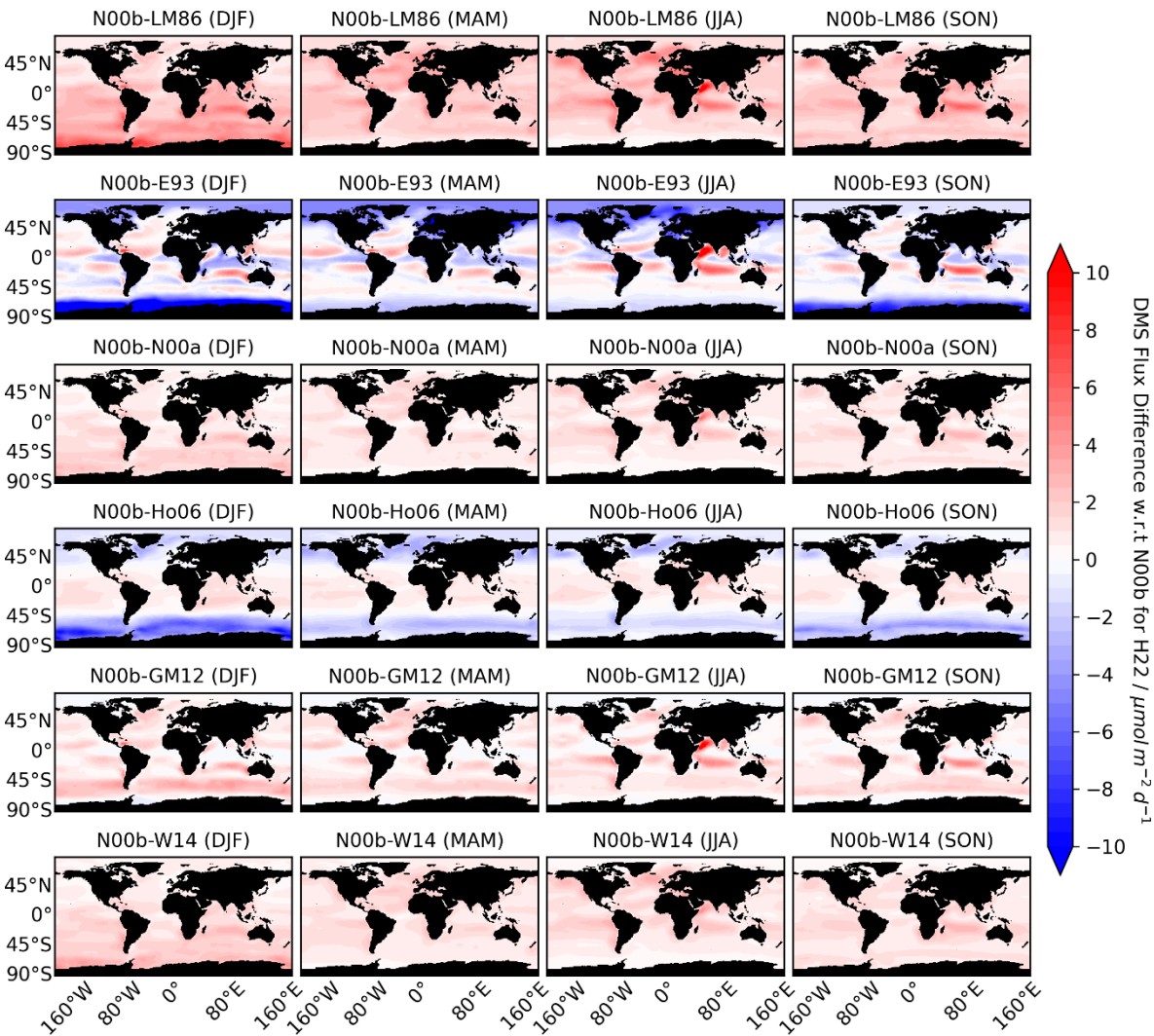

**Figure 3:** Differences between the DMS fluxes estimated using H22 with the N00b parameterization and the other seven parameterizations. For all the seasons (December-January-February (DJF),March-April-May (MAM), June-July-August (JJA), September-October-November (SON)), N00b-LM86 shows a positive difference, while the other parameterizations (E93, Ho06) show negative differences in the Southern Ocean and Arctic region, although some positive differences are also present in E93 and Ho06 in mid latitude regions. GM12, W14 and N00a show small positive differences with N00b while N00b-LM86 shows notable large positive difference. The summary of the differences in different oceanic regions is listed in Table S2.

585

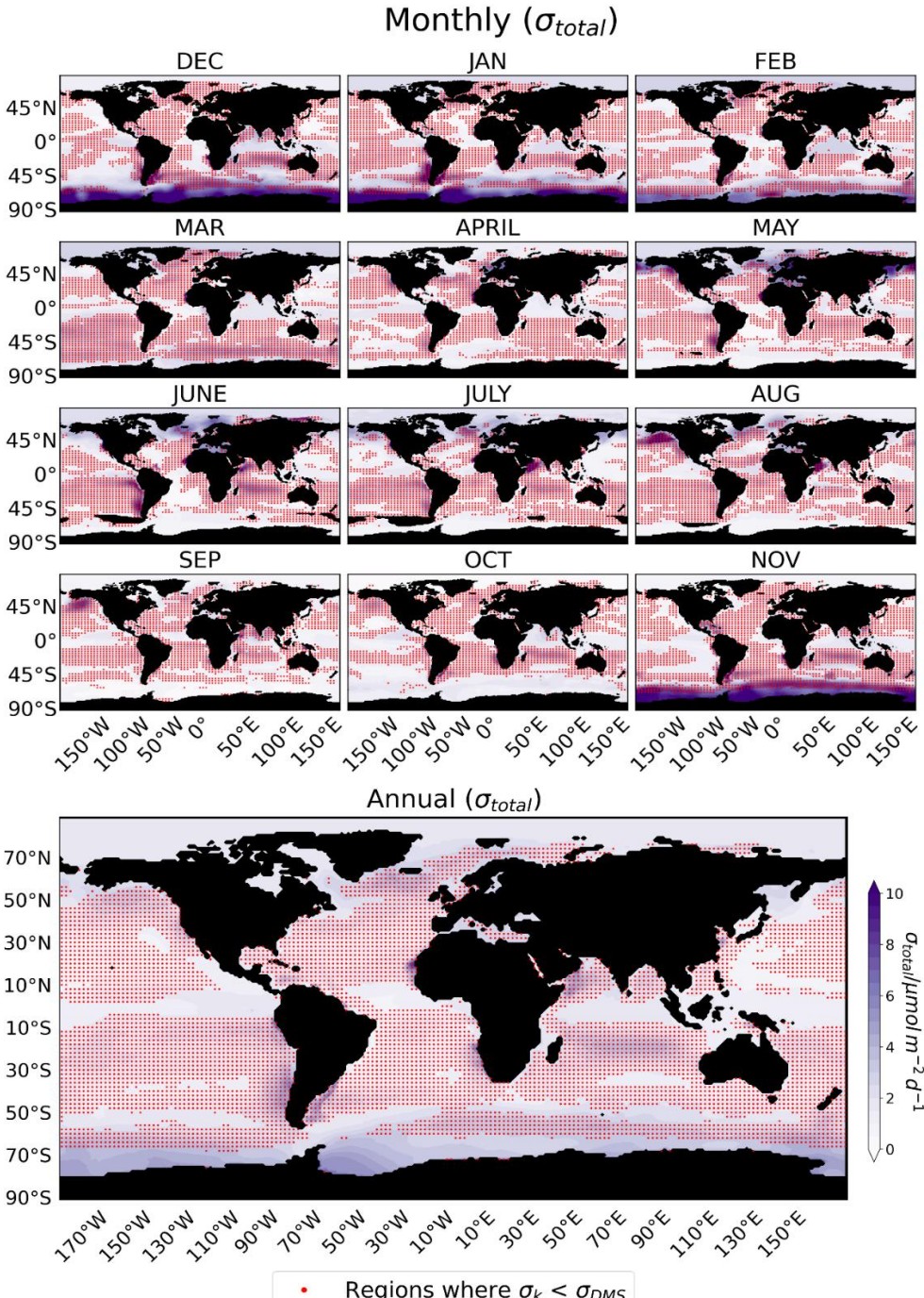

**Figure 4:** An estimate of the total variation ($\sigma_{total}$) in the flux emission, which is shown as a background map and is obtained from the standard deviations in the seawater DMS concentrations ($\sigma_{DMS}$), standard deviations in the coefficients of parametrizations ($\sigma_k$) and variation due to wind speed ($\sigma_{wind}$). $\sigma_{wind}$ has a small contribution compared to $\sigma_{DMS}$ and $\sigma_k$ (Table 1). The regions where seawater DMS concentrations drive the uncertainty are indicated by red dots ($\sigma_{DMS} > \sigma_k$), while in the other areas (no red dots), it is driven by the variation due to the choice of the flux parameterization ($\sigma_{DMS} < \sigma_k$). The maximum values of $\sigma_{total}$ is listed in Table S3.

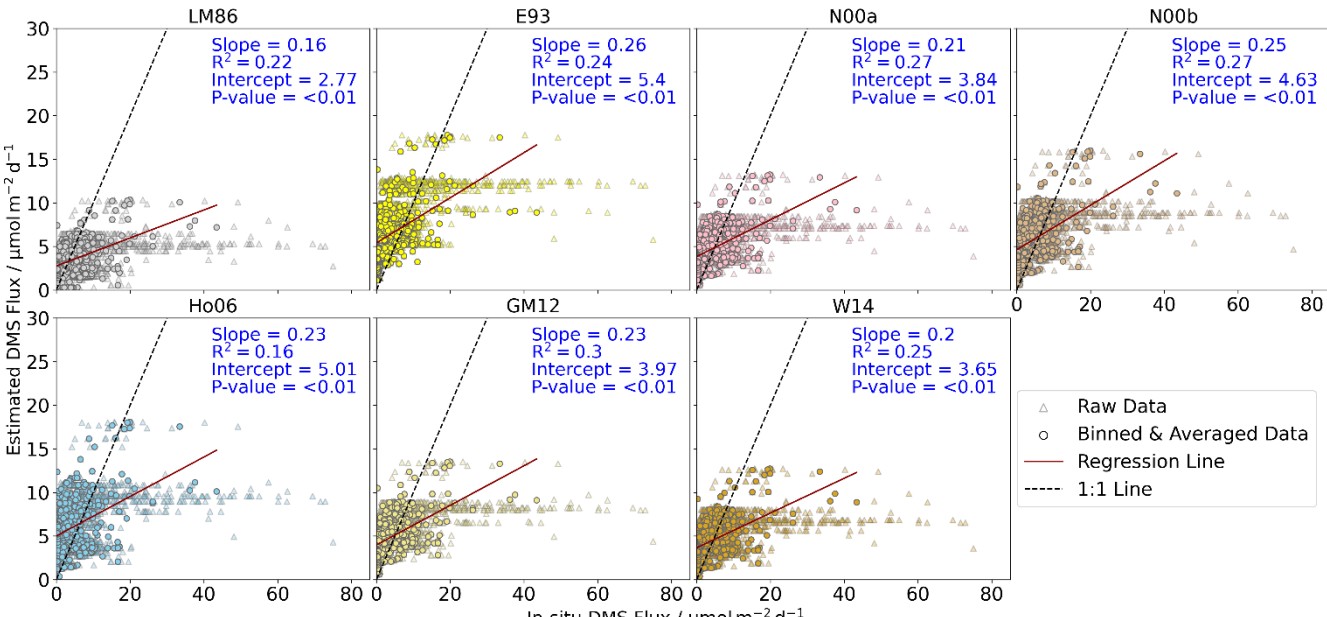

**Figure 5:** Comparison of in situ and estimated DMS fluxes (using H22) with the different parameterizations. Here, the regression analysis is done with binned and averaged in situ data at 1°×1° resolution, as the flux climatologies are also at the same resolution. The analysis shows that flux calculations result in higher fluxes than observations at low levels ($< 20\,\mu$mol m$^{-2}$ d$^{-1}$) and lower fluxes than observations at higher levels ($> 20\,\mu$mol m$^{-2}$ d$^{-1}$),which indicates that flux parametrization methods fail to represent the range accurately. The black dash line is the 1:1 representation between in situ and the estimated DMS flux points, and the dark red line is the regression line. A list of the in situ observations used for the comparison is given in Table S1.

600

