# Peer review of "Dimethyl sulfide (DMS) climatologies, fluxes, and trends - Part B: Seaair fluxes"

_EGUsphere, 2024_

## Referee Comment (RC1)

[referee-annotated manuscript omitted]

---

## Referee Comment (RC2)

**General comments:**

The manuscript provides a comparison of DMS emission field using three seawater DMS climatology and seven gas exchange parameterizations. The author then discusses the contribution of the differences in seawater DMS, gas exchange velocity, and wind speed to the DMS emission flux. Finally, the monthly emission estimates are validated using in-situ flux measurements. While DMS emission is an important topic and the results are useful, there are some issues that should be addressed to enhance its usefulness to the community.

**Specific comments:**

L25: The ocean is the dominant source of global DMS emissions. However, DMS has also been found to be emitted from vegetation on land (e.g., (Vettikkat, et al. 2020)). The author should explore the literature and list the emission sources on land, which will give readers a broader perspective on DMS emissions across the Earth's surface.

L58: Some other studies (e.g., (Blomquist, et al. 2017)) use equations which consider the bubble injection by breaking waves. Could you show some results using such equations and discuss the differences?

L64: Please add a reference.

L76: Most of the transfer velocity parameterizations in this manuscript use transfer velocities measured for gases other than DMS. However, there are parameterizations derived directly from wind speed and DMS measurements (e.g., (Yang, et al. 2011)). It would be interesting to show the results using this kind of parameterization.

L130: (Wanninkhof 2014) should be cited here.

L132: In (Wanninkhof 2014), the gas transfer velocity is given as:

$$k = 0.251 \times \langle U^2 \rangle \times (Sc/660)^{-0.5}$$

where, $\langle U^2 \rangle$ is the average of neutral stability winds at 10-m height squared, or the second moment. In the manuscript, the author uses monthly averaged wind speed. However, the difference between the two and the associated uncertainty in DMS emissions is not discussed.

L139-L145: What does "The flux due to $\sigma_{DMS}$" mean? Please also correct similar expressions as they are confusing.

The title of the manuscript is "Dimethyl sulfide (DMS) climatologies, fluxes, and trends – Part B: Sea-air fluxes", but no trend analysis is performed in this study. In L53, you mentioned that emission during 1948-2022 were used to calculate the DMS emission flux. However, no trend analysis is performed on the DMS emissions.

Introduction section: The author should add more on the sources and sinks of DMS in the ocean, and the chemical processes after it is released into the atmosphere. Then explain why DMS can affect climate.

In addition to Table S1, a figure should be added in the main text to show the locations of the in-situ measurements used for DMS flux validation, with a legend showing two methods: eddy covariance and gradient flux technique.

In Supplement, a figure should be added to show the locations of in-situ seawater measurements used to create the three seawater DMS climatologies (G18, W20, H22). This helps to determine in which regions the seawater concentrations in the climatology are more confident.

**References**

Blomquist, B. W., S. E. Brumer, C. W. Fairall, B. J. Huebert, C. J. Zappa, I. M. Brooks, M. Yang, et al. 2017. "Wind Speed and Sea State Dependencies of Air-Sea Gas Transfer: Results From the High Wind Speed Gas Exchange Study (HiWinGS)." *Journal of Geophysical Research: Oceans* 122 (10): 8034-8062.

Vettikkat, L., V. Sinha, S. Datta, A. Kumar, H. Hakkim, P. Yadav, and B. Sinha. 2020. "Significant emissions of dimethyl sulfide and monoterpenes by big-leaf mahogany trees: discovery of a missing dimethyl sulfide source to the atmospheric environment." *Atmospheric Chemistry and Physics* 20 (1): 375-389.

Wanninkhof, R. 2014. "Relationship between wind speed and gas exchange over the ocean revisited." *Limnology and Oceanography Methods* 12 (6): 351-362.

Yang, M., B. W. Blomquist, C. W. Fairall, S. D. Archer, and B. J. Huebert. 2011. "Air-sea exchange of dimethylsulfide in the Southern Ocean: Measurements from SO GasEx compared to temperate and tropical regions." *Journal of Geophysical Research* 116 (C4): c00f05.

---

## Author Comment (AC1)

**Response to Reviewer 3 comments for manuscript ID egusphere-2024-175. The comments are given in an italic typeface, and the responses are given in a bold typeface. The corresponding changes in the revised manuscript are highlighted in red.**

*3.1) This study led by D. Joge offers a valuable comparison of DMS flux parameterizations. Here are some additional thoughts to complement the two existing reviews.*

*In the introduction, it would be beneficial to clarify the distinctive contribution of this analysis. The authors briefly reference a previous intercomparison (lines 31–37) before delving into the specifics of their current work (lines 38–45). Inserting an intermediate paragraph summarizing the key differences between this study and prior research, along with the main outcomes from the companion paper (part A), would enhance the paper's coherence.*

**Response: The main outcomes of Part A are added to the introduction (L57- L59) and to the discussion section of the revised manuscript (L310 – L314).**

*3.2) Figures:*

*While the current version of the manuscript includes compelling figures, a few more could enhance reader comprehension. Here are some suggestions:*

*3.2.1) Section 2.1: Add figures to highlight the differences between the various parameterization methods (which may not be immediately clear from the equations alone). Potential figures could illustrate:*
*i) wind speed dependency of air-water gas transfer velocity for the different parameterizations, scaled to a Schmidt number at e.g., 20ºC;*
*ii) temperature dependency of air-water gas transfer velocity for the different parameterizations, scaled to different wind speeds (with one sub-figure per wind regime);*
*iii)temperature dependency of the Schmidt number for the different parameterizations.*

**Response: k vs u figures are added for all seven flux parametrizations in the supplementary text (Figs. S1 -S7), and the information related to these figures is added in the discussion section. The flux parametrization equations used in this study depend on the Schmidt number, which is a function of SST. The discussion related to this is added to this section (L316 - L320).**

*3.2.2) Section 3: While Figure 3 is commendable, Figures S3 and S4 could be more informative. A 'summary figure' combining results from these different figures could be beneficial. For instance, consider a figure where each grid box indicates the dominant contributing to the total uncertainty (using distinct colors for k, DMS, and wind). Alternatively, create one global map per parameter (k, DMS, wind) displaying, for each grid box, the percentage contribution to the total uncertainty.*

**Response: We tried to create a figure similar to the suggestion by the reviewer, but due to the coarse resolution of the data (1° x 1°), it was difficult to represent the information about uncertainty, especially in the monthly plots. Hence, we have not changed Figures S3 (now it is S10 in supplementary text) and S4 (now it is S11 in supplementary text) along with Figure 3. As the reviewer mentioned, the figures do show the needed information.**

*3.3) Additional comments:*

*3.3.1) Line 32: The statement "with the wind proven to be one of the most influencing factors" could be expanded upon. DMS flux measurements have revealed a decrease in gas transfer at medium to high wind speeds (> 10 m/s), attributed to wave-wind interactions and surfactant effects (Zavarsky et al., 2018), factors typically overlooked in traditional approaches (Bell et al., 2017). This discussion should be incorporated into the introduction.*
**Response: The text is now expanded with examples and citations (L43 - L45).**

*3.3.2) Line 43: A closing parenthesis is missing after W20.*
**Response: Parenthesis added (L56).**

*References*

*1. Bell, T. G., Landwehr, S., Miller, S. D., de Bruyn, W. J., Callaghan, A. H., Scanlon, B., Ward, B., Yang, M., and Saltzman, E. S.: Estimation of bubble-mediated air–sea gas exchange from concurrent DMS and CO2 transfer velocities at intermediate–high wind speeds, Atmospheric Chem. Phys., 17, 9019–9033, https://doi.org/10.5194/acp-17-9019-2017, 2017.*

*2. Zavarsky, A., Goddijn-Murphy, L., Steinhoff, T., and Marandino, C. A.: Bubble-Mediated Gas Transfer and Gas Transfer Suppression of DMS and CO2, J. Geophys. Res. Atmospheres, 123, 6624–6647, https://doi.org/10.1029/2017JD028071, 2018*

**Response: All the above references are cited in the revised manuscript.**

---

## Author Comment (AC2)

**Response to Reviewer 1 comments for manuscript ID egusphere-2024-175. The comments are given in an italic typeface, and the responses are given in a bold typeface. The corresponding changes in the revised manuscript are highlighted in red.**

*1.1)    Review: Dimethyl sulfide (DMS) climatologies, fluxes, and trends - Part B: Sea air fluxes By Sankirna D. Joge et al. The paper evaluated DMS fluxes based on different DMS climatologies, windspeed fields and gas transfer velocity parameterisations. The paper further evaluates which of those influences is contributing most to the differences in total flux. While the study is generally providing some useful insights, I would recommend some major revisions to the paper. In particular I am missing a proper discussion section, including comparisons to earlier comparisons among k-parameterisations and DMS fluxes. What is new/different here compared to earlier ones, are there new scientific insights/messages? Furthermore, the scale differences (temporal and spatial) need to be better discussed with respect to the in situ vs calculated fluxes. It is questionable that those are comparable in this study. The writing is generally ok, but some English language editing particularly with respect to missing articles is recommended. (I provided annotations in the pdf)*

**Response: We thank the reviewer for the interest and the thorough review of the manuscript and also for providing the annotations in the article, which have been included in the revised manuscript. The comments given in the pdf have also been answered. Answers to the detailed comments are given one by one in this document.**

*1.2)    **Detailed comments** (smaller notes and language edits are only in the pdf annotations)*

*1.2.1) The introduction has generally rather old references and could do with some updates. Especially given the discussions following Quinn and Bates 2011, maybe some newer refs would help to emphasize DMS studies are still valuable.*

**Response: The introduction section is updated with text (L25 – L40, L42 – L45), and new references have been cited.**

*1.2.2) L63: is Ca ignored here? Maybe see Steiner and Denman, 2008 (https://doi.org/10.1016/j.dsr.2008.02.010, their Fig 6) on difference in flux if the atmospheric concentration is ignored in higher emission regions, for example station papa (note dependence on boundary layer)*

**Response:  Yes, the reviewer is right that we have ignored the impact of Ca, which can play an important role in determining the total flux. However, considering the fluxes we report here are for chemistry transport models which do not include the impact of Ca, the comparison between the parameterisations is valid. We have clarified this in the revised manuscript (L80 – L83).**

*1.2.3) Section 2: Please clarify that you are not comparing flux calculations but parameterisations for the gas exchange velocity k (correct throughout). Also please add a k vs u figure and discuss the differences among parameterisations.*

**Response:  This is now explicitly mentioned (L170– L172). The k vs u plots are also added in the supplementary text (Figs. S1 – S7).**

*1.2.4) Please be accurate with the wording. The word observed should be used for actual observations in the field, not what is seen in a parameterization-derived figure. Please correct throughout as it is confusing.*

**Response: The changes have been made in the manuscript throughout as per the reviewer's suggestion.**

*1.2.5) If talking about sensitivity, ensure the sensitivity has been defined and tested, if not use a different word. Similarly, if using the word significant, a significance test needs to be made. If not, please use another word. (Locations indicated in annotated pdf).*
**Response: Corrected throughout the manuscript.**

*1.2.6) L186 ...seen in the LM86 parameterization, which consistently displays lower values than the N00b parametrization". This is to be expected if linear versus quadratic windspeed dependence is applied – should be discussed in context of a k-u figure.*
**Response: Yes, the reviewer is right in this surmise. A discussion on why LM86 shows lower values than N00b is now included (L230 – L231).**

*1.2.7) As a general rule, other than continents or large ocean basins, please identify specific locations in the map, e.g., Mauritius, Somalia... (so the reader doesn't have to take out an atlas to follow the discussion)*
**Response: The lines about maximum values and their respective location are removed from the text, and a Table S2 has been added in the supplementary text to make it easier to read.**

*1.2.8) L258 ...Overall, the choice of seawater DMS estimation method has larger influence on sea-air DMS flux than the choice of flux parameterization (Bhatti et al., 2023). Is this a result from Bahtti et al or from this study, if the latter clarify that this is also highlighted/shown in Bhatti et al. if it is not a result from this study it should be in the intro or discussion.*
**Response: The results are from this study, which can also be seen in Fig. 3. The study by Bhatti et al. (2023) is cited to support our result, along with Tesdal et al. (2016) (L281).**

*1.2.9) A discussion session is missing. While some components of the result section can be moved into the discussion session, I am missing a comparison to earlier parameterisation intercomparisons, especially Tesdal et al 2015, https://doi.org/10.1071/EN14255 , but also e.g., Steiner et al. https://elischolar.library.yale.edu/journal_of_marine_research/170 their Fig 1. Also, some discussion on why and where the parameterisations differ - link to k versus u figure. Improved discussion on spatial and temporal scales in context with the comparison to in situ observations.*
**Response: A detailed discussion is added in Section 4, and the points suggested by the reviewer are now included in this section (L305 - L342).**

*1.2.10) Results section and table 1: The text with the long lists of max and mins is rather tedious to read. Maybe remove a good part of it and add a max column into Table 1 or add a table with max/mean/mins in the Appendix, potentially divided into different ocean basins.*
**Response: As suggested by the reviewer, tables S2 and S3 have been added to the supplementary text, and the detailed description of the results has been removed.**

[revised manuscript text omitted]

---

## Author Comment (AC3)

**Response to Reviewer 2 comments for manuscript ID egusphere-2024-175. The comments are given in an italic typeface, and the responses are given in a bold typeface. The corresponding changes in the revised manuscript are highlighted in red.**

*2.1) General comments:*

*The manuscript provides a comparison of DMS emission field using three seawater DMS climatology and seven gas exchange parameterizations. The author then discusses the contribution of the differences in seawater DMS, gas exchange velocity, and wind speed to the DMS emission flux. Finally, the monthly emission estimates are validated using in-situ flux measurements. While DMS emission is an important topic and the results are useful, there are some issues that should be addressed to enhance its usefulness to the community.*

**Response: We thank the reviewer for thoroughly reviewing the manuscript. The answers to the detailed comments are given below.**

*2.2) Specific comments:*

*2.1.1) L25: The ocean is the dominant source of global DMS emissions. However, DMS has also been found to be emitted from vegetation on land (e.g., (Vettikkat, et al. 2020)). The author should explore the literature and list the emission sources on land, which will give readers a broader perspective on DMS emissions across the Earth's surface.*

**Response: Yes, a line related to the DMS sources from vegetation on land is added and a reason why the ocean is a dominant source of DMS is given along with the relevant references (L36 – L39).**

*2.1.2) L58: Some other studies (e.g., (Blomquist, et al. 2017)) use equations which consider the bubble injection by breaking waves. Could you show some results using such equations and discuss the differences?*

**Response: Bubble bursting is an important process, but its impact on DMS emissions is not clear. It does affect sea salt aerosol formation, but describing these other processes, which are not directly relevant to DMS emissions, is beyond the scope of our topic.**

*2.1.3) L64: Please add a reference.*
**Response: Added (L85).**

*2.1.4) L76: Most of the transfer velocity parameterizations in this manuscript use transfer velocities measured for gases other than DMS. However, there are parameterizations derived directly from wind speed and DMS measurements (e.g., (Yang, et al. 2011)). It would be interesting to show the results using this kind of parameterization.*

**Response: Section 2.1.5 contains the Goddijn-Murphy et al. parametrization (GM12), which is a synthesis of $k_{DMS}$ using (at that time) all available eddy covariance DMS flux observations. The data from Yang et. al. (2011) is included in this synthesis.**

*2.1.5) L130: (Wanninkhof 2014) should be cited here.*
**Response: Now included (L163).**

*2.1.6) L132: In (Wanninkhof 2014), the gas transfer velocity is given as: $k=0.251 \times \langle U2 \rangle \times (Sc/660)^{-0.5}$ where, $\langle U2 \rangle$ is the average of neutral stability winds at 10-m height squared, or the second moment. In the manuscript, the author uses monthly averaged*

*wind speed. However, the difference between the two and the associated uncertainty in DMS emissions is not discussed.*

**Response: The discussion regarding the difference and its associated uncertainty is now in section 4 –'*The gas transfer velocity equation of W14 uses the square of the average neutral stability winds at 10 m height or second moment i.e., average of the quadratic windspeed. In this study, we used monthly average windspeed, i.e., the quadratic average windspeed for W14. The first method of calculation will estimate higher k values than the second one due to the averaging of the winds. We checked the differences between the two and found that the maximum difference is not more than 4.3 cm h$^{-1}$ for June, July and August months and it is less than 2 cm h$^{-1}$ for rest of the year, which does not contribute pointedly to the large uncertainty.*'.**

*2.1.7) L139-L145: What does "The flux due to $\sigma$DMS" mean? Please also correct similar expressions as they are confusing.*

**Response: $\sigma_{DMS}$ is calculated by calculating the standard deviation between H22, W20 and G18. This $\sigma_{DMS}$ is used along with N00a parametrization, windspeed and SST data to get standard deviation in flux, which is shown in the monthly and annual $\sigma_{DMS}$ plots (Fig. S10)." This standard deviation in calculated flux is the flux due to $\sigma_{DMS}$. The similar expressions are corrected for the other standard deviation parameters. This is now explained in the revised manuscript (L178 - L185).**

*2.1.8) The title of the manuscript is "Dimethyl sulfide (DMS) climatologies, fluxes, and trends – Part B: Sea-air fluxes", but no trend analysis is performed in this study. In L53, you mentioned that emission during 1948-2022 were used to calculate the DMS emission flux. However, no trend analysis is performed on the DMS emissions.*

**Response: Trend analysis is performed for DMS seawater concentration in Joge:Part A. An increasing trend of seawater DMS concentration is obtained, which also indicates that sea-air flux will increase. In this manuscript, Part B, the focus is on sea-air flux parametrization.**

*2.1.9) Introduction section: The author should add more on the sources and sinks of DMS in the ocean, and the chemical processes after it is released into the atmosphere. Then explain why DMS can affect climate.*

**Response: This information is now added in the introduction section (L25 - L35).**

*2.1.10) In addition to Table S1, a figure should be added in the main text to show the locations of the in-situ measurements used for DMS flux validation, with a legend showing two methods: eddy covariance and gradient flux technique.*

**Response: The suggested figure is added in the supplementary text as Figure S12. The text is also added in the data and methodology section (L70 - L72).**

*2.1.11) In Supplement, a figure should be added to show the locations of in-situ seawater measurements used to create the three seawater DMS climatologies (G18, W20, H22). This helps to determine in which regions the seawater concentrations in the climatology are more confident.*

**Response: This figure is available in part A of this manuscript.**

*References*
1. *Blomquist, B. W., S. E. Brumer, C. W. Fairall, B. J. Huebert, C. J. Zappa, I. M. Brooks, M. Yang, et al. 2017. "Wind Speed and Sea State Dependencies of Air-Sea Gas Transfer: Results from the High Wind Speed Gas Exchange Study (HiWinGS)." Journal of Geophysical Research: Oceans 122 (10): 8034-8062.*

2. *Vettikkat, L., V. Sinha, S. Datta, A. Kumar, H. Hakkim, P. Yadav, and B. Sinha. 2020. "Significant emissions of dimethyl sulfide and monoterpenes by big-leaf mahogany trees: discovery of a missing dimethyl sulfide source to the atmospheric environment." Atmospheric Chemistry and Physics 20 (1): 375-389.*

3. *Wanninkhof, R. 2014. "Relationship between wind speed and gas exchange over the ocean revisited." Limnology and Oceanography Methods 12 (6): 351-362.*

4. *Yang, M., B. W. Blomquist, C. W. Fairall, S. D. Archer, and B. J. Huebert. 2011. "Air-sea exchange of dimethylsulfide in the Southern Ocean: Measurements from SO GasEx compared to temperate and tropical regions." Journal of Geophysical Research 116 (C4): c00f05.*

**Response: All the above references are cited in the revised manuscript.**

---

## Referee Report (RR1)

**General comments:**

The authors have made significant efforts to address my concerns and improve the manuscript. However, a few critical parts still require attention before the manuscript can be accepted for publication.

**Specific comments:**

1) Trend analysis is necessary to match the manuscript with the title. The DMS emission flux is theoretically determined by the seawater concentrations and gas transfer parameterization. Environmental factors, such as wind speed, affect the gas transfer rate. While the manuscript highlights the discrepancies in seawater concentrations as the main contributor to uncertainty in global emission flux, it is crucial to present spatial variations in emission flux trends or at least global emission trends. This is important due to the role of DMS in climate, and incorporating this analysis would greatly enhance the contribution of this work.

2) Reviewer 3 provided excellent suggestions for a figure to illustrate the differences between parameterizations. Although the authors have included Figure S1-S7 using monthly data, it remains difficult for readers to follow. I recommend adding a figure in the main text with three subfigures: the first subfigure should show the wind speed dependence of the gas transfer rate; the second subfigure should display the temperature dependence; the third should illustrate the temperature dependence of the Schmidt number (Sc). The figure should be derived from theoretical estimates, and monthly data are not needed for their creation.

---

## Author Response (AR2)

**Response to Reviewer 2 comments for manuscript ID egusphere-2024-175. The comments are given in an italic typeface, and the responses are given in a bold typeface. The corresponding changes in the revised manuscript are highlighted in red.**

*2.1) General comments:*

*The authors have made significant efforts to address my concerns and improve the manuscript. However, a few critical parts still require attention before the manuscript can be accepted for publication.*

**Response : We thank the reviewer for thoroughly reviewing the manuscript again. The specific comments given by reviewer are answered one by one and the corresponding changes are made in the manuscript.**

**Specific comments:**

*2.1.1) Trend analysis is necessary to match the manuscript with the title. The DMS emission flux is theoretically determined by the seawater concentrations and gas transfer parameterization. Environmental factors, such as wind speed, affect the gas transfer rate. While the manuscript highlights the discrepancies in seawater concentrations as the main contributor to uncertainty in global emission flux, it is crucial to present spatial variations in emission flux trends or at least global emission trends. This is important due to the role of DMS in climate, and incorporating this analysis would greatly enhance the contribution of this work.*

**Response: New figures related to the flux trend analysis are now added in the supplementary text (Figures S8 and S9). The following text has also been added in the discussion section (L329 – L333):**

**"From 1998 to 2010, both G18 and W20 show an increasing trend in seawater DMS (Joge: Part A). Using the calculated seawater DMS concentrations, G18 and W20 DMS flux trends are also calculated for each parameterization method using the bootstrap resampling method (explained in Part A). The DMS flux trend also shows an increase for all the parameterizations (Figures S8 and S9). DMS flux values are between 3 $\mu$mol m$^{-2}$ d$^{-1}$ and 6 $\mu$mol m$^{-2}$ d$^{-1}$ for E93 and Ho06, but lower in LM86. GM12 and W14 show a similar range while N00b shows a larger range compared to N00a."**

[Figure]

**Figure S8 :** DMS flux trend using seawater DMS concentrations of G18 for each parameterization method. Trend is calculated using bootstrap resampling method. The trend is significant if $t_B$ > 2.

[Figure]

**Figure S9 :** DMS flux trend using seawater DMS concentrations of W20 for each parameterization method. Trend is calculated using bootstrap resampling method. The trend is significant if $t_B$ > 2.

*2.1.2) Reviewer 3 provided excellent suggestions for a figure to illustrate the differences between parameterizations. Although the authors have included Figure S1-S7 using monthly data, it remains difficult for readers to follow. I recommend adding a figure in the main text with three subfigures: the first subfigure should show the wind speed dependence of the gas transfer rate; the second subfigure should display the temperature dependence; the third should illustrate the temperature dependence of the Schmidt number (Sc). The figure should be derived from theoretical estimates, and monthly data are not needed for their creation.*

**Response: A new figure is added in the main text (Figure 1) as suggested by reviewer 3, and the following text is also added at the end of section 2.1 (L157 – L165).**

**"$Sc_{n00a}$, $Sc_{n00b}$ and $Sc_{gm12}$ use the same formulation as that of $Sc_{lm86}$. LM86 shows three linear regions when compariong $k$ vs $u$, as defined by Eq.1-3. GM12 shows a linear dependency on the**

windspeed, while other formulations show a nonlinear dependency (Figure 1a). When the temperature is changed from -2 °C to 33.2 °C, then there is am increase and a spread between the k values. This is due to the temperature dependence of *Sc*, which nonlinearly decreases with temperature (Figure 1b). *Sc* is the ratio of kinematic viscosity ($v$) and molecular diffusivity (D) i.e., *Sc* = $v$/D (Liss and Merlivat, 1986). So, as *Sc* decreases with temperature, the molecular diffusion rate increases from higher concentrations (seawater) to lower concentrations (atmosphere) and hence the value of *k* increases with windspeed (Figure 1). Note that even though Ho06 is independent to the *Sc*, there is a small spread in the values of *k* with temperature. This is due to the Ostwald solubility coefficient for DMS according to McGillis et al. (2000) used in Ho06, and this coefficient has a temperature dependence."

[Figure]

**Figure 1: (a)** Coefficient of gas transfer velocity ($k$) vs windspeed at 10 m from sea surface ($U_{10}$) at constant temperature (20 °C) and at different temperature values (-2 °C to 33.2 °C). **(b)** Schmidt number (*Sc*) vs temperature (T) for each flux parameterization methods. $Sc_{n00a}$, $Sc_{n00b}$ and $Sc_{gm12}$ has same equation as that of $Sc_{lm86}$. $Sc_{Rn}$ is the Schmidt number for radon used in E93 parameterization. Schmidt number (*Sc*) decreases with increase in temperature and hence gas transfer coefficient ($k$) increases.